

# Comparison of formaldehyde measurements by Hantzsch, CRDS and DOAS in the SAPHIR chamber

Marvin Glowania[1], Franz Rohrer[1], Hans-Peter Dorn[1], Andreas Hofzumahaus[1], Frank Holland[1], Astrid Kiendler-Scharr[1], Andreas Wahner[1], and Hendrik Fuchs[1]

[1]Institute of Energy and Climate Research, IEK-8: Troposphere, Forschungszentrum Jülich GmbH, Jülich, Germany

**Correspondence:** H. Fuchs (h.fuchs@fz-juelich.de)

**Abstract.**

Three instruments using different techniques measuring gaseous formaldehyde (HCHO) concentrations were compared in experiments in the atmospheric simulation chamber SAPHIR at Forschungszentrum Jülich. One instrument detected HCHO by using the wet-chemical Hantzsch reaction for efficient gas-phase stripping, chemical conversion and fluorescence measure-

ment (AL4021, Aero Laser GmbH). An internal permeation HCHO source allows for daily calibrations. It was characterized by sulfuric acid titration (overall accuracy 8.5 %). Measurements have a time resolution of 90 s with a limit of detection ($3\,\sigma$) of 0.3 ppbv. In addition, a new commercial instrument making use of cavity ring-down spectroscopy (CRDS) determined concentrations of HCHO, water, and methane (G2307, Picarro Inc.). The limit of detection ($3\,\sigma$) is specified as 0.3 ppbv for an integration time of 300 s and the accuracy is limited by the drift of the zero signal (manufacturer specification 1.5 ppbv).

A custom-built, high-resolution laser differential optical absorption spectroscopy (DOAS) instrument provided HCHO measurements with a limit of detection ($3\,\sigma$) of 0.9 ppbv and an accuracy of 6 % using an optical multiple reflection cell. The measurements were conducted from June to December 2019 in experiments in which either ambient air was flowed through the chamber or the photochemical degradation of organic compounds in synthetic air was investigated. Measured HCHO concentrations were up to 8 ppbv. Various mixtures of organic compounds, water vapour, nitrogen oxides, and ozone concen-

trations were present in these experiments. Results demonstrate the need to correct the baseline in the measurements of the Hantzsch instrument to compensate for drifting background signals. Corrections were equivalent to HCHO mixing ratios in the range of 0.5 to 1.5 ppbv. The baseline of the CRDS instrument showed a linear dependence on the water-vapour mixing ratio with different slopes of $(-11.20 \pm 1.60)$ ppbv %$^{-1}$ and $(-0.72 \pm 0.08)$ ppbv %$^{-1}$ above and below 0.2 % water vapour mixing ratio, respectively. In addition, the intercept of these linear relationships drifted with time within the specification of the

instrument (1.5 ppbv), but appeared to be equal for all water mixing ratios. Regular zero measurements are required to account for the changes in the instrument zero. After correcting for the baselines of measurements by the Hantzsch and the CRDS instruments, a linear regression analysis of measurements from all three instruments in experiments with ambient air results in a good agreement with slopes between 0.93 and 1.07 with negligible intercepts (linear correlation coefficients $R^2 > 0.96$). The new, small-sized CRDS instrument measures HCHO with a good precision and is accurate, if the instrument zero is taken

into account. Therefore, it can provide accurate and calibration-free measurements like the DOAS instrument with a slightly reduced precision compared to the Hantzsch instrument.



## 1 Introduction

Formaldehyde (HCHO) is a pollutant that is present in ambient air but also indoors. It affects human health by irritations of the respiratory system and by being carcinogenic (Gupta et al., 1982; Casset et al., 2005; World Health Organization, 2010; Fortems-Cheiney et al., 2012; Liu et al., 2015; Salthammer, 2019; Qin et al., 2020). It is formed in the atmosphere as product of the oxidation of volatile organic compounds (VOCs) and in combustion processes including biomass burning. HCHO is also directly emitted by anthropogenic activities (Parrish et al., 2012). The largest chemical production is globally from the oxidation of methane ($CH_4$) ($\sim 970 \, \mathrm{Tg \, yr^{-1}}$) and from non-methane volatile organic compounds (NMVOCs) such as terpenes and other hydrocarbons ($\sim 250 \, \mathrm{Tg \, yr^{-1}}$) from biogenic sources (Anderson et al., 2017; Zhang et al., 2019). The removal of HCHO from the troposphere is mainly determined by photolysis and the oxidation by hydroxyl radicals (OH) forming hydroperoxy radicals ($HO_2$) and carbon monoxide. Therefore, photolysis of HCHO is as a significant source for radicals in the troposphere and contributes to the oxidation capacity of the troposphere. Subsequent radical-radical recombination reactions can be an important source for hydroperoxides in the troposphere. This is specifically of importance, if the reaction of $HO_2$ with nitric oxide (NO) cannot compete like typically found in remote marine regions. HCHO can also removed by dry and wet surface deposition. The efficiency of these processes depends on the nature of the surface, but is globally a small sink for HCHO (Anderson et al., 2017; Alvarado et al., 2019; Wolfe et al., 2019). Overall, HCHO has a typical chemical lifetime in the troposphere of a few hours during the day and up to two days during night, when photolysis and the reaction with OH do not play a big role (Lowe and Schmidt, 1983; Seinfeld and Pandis, 2016).

Formaldehyde mixing ratios in the lower troposphere range between 200 to 500 parts per trillion by volume (pptv) in marine and remote areas, but can be up to multiple parts per billion by volume (ppbv) in urban environments, where sources for HCHO are highest (Still et al., 2005; Nogueira et al., 2014; Li et al., 2014).

The total yield of HCHO in the oxidation of complex organic compounds is often uncertain, because HCHO can be the product of numerous chemical reactions (Kuhn et al., 2007; Salthammer, 2019). Chemical models are often not capable of reproducing HCHO concentrations in the atmosphere and of HCHO columns derived from satellite observations. The comparison exercise "Chemistry Climate Model Initiative" (Anderson et al., 2017), for example, showed that current global models underestimate HCHO columns by 4 to 50 %. Chamber and laboratory experiments contribute profoundly to the development of chemical oxidation mechanisms under controlled conditions (Jianyin Xiong et al., 2011; He et al., 2019; Novelli et al., 2020). In order to determine HCHO yields in these experiments, accurate and precise measurements of HCHO are needed.

Strong absorption lines in the UV region allow for a sensitive detection of formaldehyde by differential absorption (DOAS) instruments (Dorn et al., 1995; Platt and Stutz , 2007) and cavity-based absorption spectroscopy (Washenfelder et al., 2016; Liu et al., 2019). Detection by absorption in the IR region is also possible by either Fourier transformation infra-red spectroscopy (FTIR), tunable diode laser spectroscopy (TDL) (Weibring et al., 2007; Shutter et al., 2019), quantum cascade laser spectroscopy (QCLS) (McManus et al., 2010) and cavity ring-down spectroscopy (Picarro Inc.). Absorption spectroscopy has the advantage of being calibration-free and the absorption cross sections of HCHO are high enough to reach limits of detection within the range of a few hundred parts per trillion per volume (pptv) mixing ratios of formaldehyde which is sufficiently low



for many applications. Most of the instruments are custom-built and require pre-knowledge of operators, but there are now two commercially available instruments (CRDS by Picarro Inc. and TDLS by Aeros Technology).

Compared to instrument making use of absorption spectroscopy, a lower limit of detection of less than 35 pptv is achieved by laser-induced fluorescence (LIF) detection after excitation at 353nm (Hottle et al., 2009; Kaiser et al., 2014). Calibration is achieved by using a gas standard or by a portable permeation source that are regularly checked by FTIR spectroscopy. LIF instruments are also custom-built including the fibre laser needed for the excitation.

Wet chemistry methods are widely used to detect formaldehyde. Sampling with cartridges for derivatization with 2,4-dinitrophenylhydrazine (DNPH) and subsequent offline analysis with high-performance liquid chromatography (HPLC) has a low limit of detection (40 pptv), but requires comparably high experimental effort (Winberry et al., 1999). Due to the long sampling time of typically 1 hour the time resolution is less than spectroscopic methods. Another online wet-chemical method is based on the Hantzsch reaction, in which aqueous formaldehyde reacts with acetylacetone (Kelly and Fortune, 1994). The concentration of the product (3,5-diacetyl-1,4-dihydrolutidine) is then measured by fluorescence after excitation at 410 nm. This type of instrument is nowadays commercially available by Aero Laser GmbH. A low limit of detection of 100 pptv is reached at a high time resolution of minutes. The disadvantage compared to spectroscopic methods are the need for regular maintenance work and calibrations with liquid and gaseous standards (see below) and the consumption of hazardous liquids.

Proton-transfer mass-spectrometry (PTR-MS) can also detect formaldehyde but due to the low proton affinity the sensitivity is not satisfying and the sensitivity is low and exhibits a strong dependence on the water vapour mixing ratio(Vlasenko et al., 2010; Warneke et al., 2011; Yuan et al., 2017). Therefore, formaldehyde concentrations have not become a standard measurement of PTR-MS instruments. A direct comparison measurement of instruments using different techniques, calibration and data evaluation procedures is a well approved way to evaluate the quality of the data and to identify possible instrumental artefacts, inaccuracy of calibration procedures or instrumental interferences (Grossmann, 2003; Inomata et al., 2008; Warneke et al., 2011).

Several comparison exercises have been performed so far. Eleven Comparisons that were done before 2005 are summarized and discussed in Hak et al. (2005). They were performed during field and chamber experiments utilizing different techniques. Most often an absorption spectrometer and Hantzsch instruments were involved. Hak et al. (2005) concluded that there is high variability in the level of agreement between measurements of instruments without exhibiting a specific pattern. Most often calibration of instruments were assumed to be the likely reason for disagreement. The authors report the comparison of four different techniques (broadband DOAS, FTIR, Hantzsch, chromatography after cartridge sampling) during measurements at an urban site. Results showed agreement of measurements by DOAS, Hantzsch and FTIR instruments within 11 %, but also strong variations in the agreement between Hantzsch measurements and other instruments. Differences between Hantzsch instruments were attributed to insufficiently working scrubbers that are used for zeroing of instruments and differences in the calibration results. For absorption instruments, Hak et al. (2005) pointed out that the use of different recommendations for (differential) absorption cross sections can lead to disagreement. In addition, the resolution of spectrometers needs to be correctly taken into account to determine the effective cross section.





In another comparison study by Wisthaler et al. (2008) that was done in the atmospheric simulation chamber SAPHIR at Forschungszentrum Jülich measurements by DOAS, DNPH-HPLC, PTR-MS, and 2 Hantzsch instruments were compared for controlled atmospheric conditions in 3 experiments. Deviations between measurements of instruments were up to 50 %. Several analytical problems were identified: (1) Measurements showed that the derivatization efficiency in the DNPH-HPLC method was significantly lowered in dry air. (2) Like in previous comparisons uncertainties in the zeroing led to a bias in the

measurements of of the Hantzsch instrument. (3) Measurements by Hantzsch were lower in the presence of ozone (45 ppbv) compared to DOAS measurements, which are likely less affected by ozone, if formaldehyde concentrations are high compared to ozone concentrations. However, deviations were variable among experiments, so that connection to ozone was uncertain.

Formaldehyde measurements by a LIF instruments were compared to Hantzsch measurements in the SAPHIR chamber in 2014 (Kaiser et al., 2014) and to a recently developed commercial TDLAS system by Aeris Technology in ambient air

(Shutter et al., 2019). The agreement between measurements by the LIF and TDLAS instruments was better than 8 % for formaldehyde mixing ratios higher than 1 ppbv. The comparison of LIF and Hantzsch instruments in SAPHIR allowed for a systematic investigation, if measurements were affected by ozone or water. No systematic deviations with the presence of water or ozone could be found, so that observations in (Wisthaler et al., 2008) with respect to a potential ozone interference in the measurements by Hantzsch could not be confirmed. Measurements between Hantzsch and LIF instruments agreed within

the combined uncertainties of calibrations (13 %).

Reports of instrument comparisons concluded that the measurement of formaldehyde remains challenging specifically for atmospheric concentrations in the low ppbv range. Calibration and instrumental zeros were identified as the major source of systematic uncertainties in the data. In this work, formaldehyde measurements from three different instruments are compared in experiments in the SAPHIR chamber located at Forschungszentrum Jülich. One instrument made use of the Hantzsch tech-

nique that was also applied in most of the previous comparisons. A high-resolution DOAS system (308 nm) for the detection of hydroxyl radical concentrations provided also formaldehyde concentrations. In addition, formaldehyde was detected by a recently available commercial CRDS instrument from Picarro Inc.. Experiments included the investigation of the photochemical oxidation of specific organic compounds as well as experiments, in which ambient air was flowed through the chamber. Therefore, these experiments gave the chance to investigate the performance of the instruments in controlled conditions that

allowed the systematic variation of parameters and for realistic ambient conditions and concentrations over a long period of time.

## 2    Experimental procedure

### 2.1    Atmospheric simulation chamber SAPHIR

Measurements were performed from June to December 2019 in experiments in the outdoor atmospheric simulation cham-

ber SAPHIR at Forschungszentrum Jülich, Germany. Detailed descriptions of the chamber and its properties can be found elsewhere (Rohrer et al., 2005).





The SAPHIR chamber consists of a cylindrical, double-wall Teflon (FEP) film with a length and diameter of 18 m and 5 m, respectively, and an effective volume of 270 m$^3$. The film is mounted in a metal frame with a shutter regulating the penetration of natural sunlight. Two fans ensure mixing of trace gases within 2 minutes. Transmittance of solar radiation through the Teflon

film of the chamber is regularly determined by actinometric experiments (Bohn and Zilken, 2005). The volume between the two Teflon films is permanently flushed with pure nitrogen (Linde, purity > 99.9999 %) and the chamber is held under slightly increased atmospheric pressure ($\sim$ 30 Pa) to prevent any contamination from ambient air. Air that is consumed by instruments or small leakages is permanently replenished by dry synthetic air. This leads to a dilution of trace gases by 3 to 5 % per hour.

The Teflon film releases small amounts of nitrous acid (HONO) and small hydrocarbons such as formaldehyde and acetalde-

hyde when it is exposed to solar radiation. The source strength depends on temperature, illumination and humidity (Rohrer et al., 2005). For formaldehyde, the source strength is approximately $(20 \pm 10)$ pptv min$^{-1}$ (30 % RH at 298 K) for clear sky summer conditions ($j(NO_2) \approx 5 \times 10^{-3}$ s$^{-1}$). This value was determined in experiments in this work, when the chamber only contained humidified clean synthetic air that was exposed to sunlight. Similar values were obtained in the past.

Some of the experiments carried out in 2019 and included in this work were part of the Jülich Atmospheric Chemistry

Project (JULIAC), which was designed to investigate the seasonal and diurnal variation of atmospheric trace gases, radicals, and particles in air influenced by anthropogenic and biogenic emissions. Ambient air was continuously sampled for four weeks in each season of the year through an inlet line made of Silconert coated stainless steel that was mounted at a 50 m high tower. Large particles (> 10 μm) were removed in a cyclone before the air was transported into the chamber. A blower compressed the air before entering the chamber (pressure difference: 15 hPa). Only a fraction (250 to 260 m$^3$ h$^{-1}$) of the total flow

(660 m$^3$ h$^{-1}$) controlled by a three-way valve flowed into the chamber. This flow rate results in a residence time of air in the chamber of approximately 1 h. Temperature, pressure, relative humidity, and solar radiation were constantly monitored inside and outside of the SAPHIR chamber. During strong wind or heavy rain the louvre system of SAPHIR was closed to prevent damage of the Teflon film.

In addition to measurements in ambient air (JULIAC), data from experiments are included in the analysis, in which the photo-

oxidation of anthropogenic and biogenic volatile organic compounds (e.g. isoprenoids, terpenes and derivatives of acetone) were investigated under controlled conditions. Before the start of a typical experiment, the chamber was flushed with a mixture of ultra-pure nitrogen and oxygen (Linde, purity > 99.99990 %) to remove any remaining trace gases or contaminations. In most of the experiments, the chamber air was humidified by evaporating Milli-Q$^{\circledR}$ water that was flushed into the chamber with a high flow of synthetic air. Organic or inorganic compounds (e.g. alkenes, nitrogen oxides, ozone) or particles were injected

to compose various conditions, for which the oxidation and degradation of organic compounds were investigated either in the dark or under solar radiation. Emissions from up to 6 trees housed in a plant-chamber (Hohaus et al., 2016) were occasionally transferred into the chamber, in order to study their photo-oxidation. Reference experiments under similar conditions and with no injections of the analytes provide background data in order to determine the strength of chamber sources e.g. for HCHO.





**Table 1.** Specifications of Hantzsch, CRDS and DOAS instruments as stated by the manufacturer or reported in previous literature.

|  | Hantzsch | CRDS | DOAS |
|---|---|---|---|
| integration time / s | 90 | 300 | 130 |
| limit of detection (1 $\sigma$) / ppbv | 0.1 | 0.1 | 0.3 |
| accuracy (1 $\sigma$) | 8.5 % | 10 % | 6 % |
| references | Aero-Laser GmbH | Picarro Inc. | Hausmann et al. (1997) |

## 2.2 HCHO detection by the wet chemical method using the Hantzsch reaction

One of the instruments for the detection of HCHO in this work is a commercial instrument (AL4021, Aero-Laser GmbH) that is based on the wet chemistry Hantzsch method. Air is sampled at a flow rate of 1 L min$^{-1}$ ($F_{gas}$) through a 5 m long 1/4 " O.D. PFA Teflon tube. Any losses or production of HCHO within the contact time ($\approx 4$ s) of the air with the Teflon surface are expected to be negligible (Sumner et al., 2001). A stripping coil is used to transfer the HCHO from the gas phase to a 50 mM sulphuric acid solution. The stripping efficiency ($\alpha$) describes the percentage of formaldehyde being physically separated from
the gaseous phase into the liquid phase inside a coil, which is installed inside a box. The stripping is nearly quantitative with an efficiency of $\alpha = (0.99 \pm 0.01)$ stated by the manufacturer (Aero-Laser GmbH). The liquid flow is controlled by a peristaltic pump at a flow rate of approximately $F_{liq} \sim (450 \pm 40)$ µL min$^{-1}$. The actual measurement of the HCHO concentration in the aqueous solution is then performed by a liquid phase chemical reaction in a reactor. Aqueous HCHO reacts with acetylacetone (2,4-pentadione) and ammonia forming pyridine dye 3,5-diactyl-1,4-dihydrolutidine (DDL) in the Hantzsch-
reaction. The pyridine dye is detected by fluorescence at 510 nm after excitation by 410 nm radiation from a UV-LED. The time response of the instrument is about 90 s (10 % to 90 %) with an additional time delay of 5 to 6 min due to the time required for the transport of the liquid from the coil to the fluorimeter. The 1-$\sigma$ limit of detection is stated as 100 pptv by the manufacturer (Aero-Laser GmbH).

For the experiments in this work, the instrument was placed in an air-conditioned sea container next to the chamber. Parts
of the instrument are temperature-stabilized: the stripping coil at 283 K, the reaction volume at 341 K and the fluorimeter at 308 K. This ensures an approximate constant reaction efficiency and fluorescence detection sensitivity (Rurack and Spieles, 2011; Resch-Genger and Rurack, 2013). The tubing of the peristaltic pumps was exchanged every two weeks due to a degradation of tubes and in order to prevent occlusions or strong loss of pumping performance, which cause changes in the instrument sensitivity of up to 10 % within this time span (Section 3.1). To prevent chemical degradation and ageing, all chemicals were
stored in a box at a controlled temperature of 277 K. Chemicals needed to be refilled every one to two weeks. Stopping the liquid flow or switching the instrument off for regular tubing exchange, cleaning or maintenance required an additional warm-up time (30 to 120 min) for the instrument to be in an thermal equilibrium and to stabilize the flow rates of liquids in order to achieve a stable response signal.





The instrument shows a significant zero signal ($S_0$) that needs to be subtracted from the total measured signal ($S$), in order

to derive the signal caused by HCHO. Automatized zero measurements are performed for about 20 minutes every two to three hours by removing HCHO from the sampled air by means of a scrubber consisting of a cylindrical glass, which is filled with a brown rod-shaped material, which — according to the manufactures — provides reliable and sufficient HCHO scavenging. The internal software of the instrument takes the last zero measurement to evaluate the current measurement.

The HCHO mixing ratio in the sampled gas (flow rate $F_{gas}$) is derived from the instrument sensitivity $E_{Cal}$. The signals are

normalized to the inverse of the flow rate of the liquid solution ($F_{liq}$). Additional parameters like the molar volume $V_{mol}$ under standard ambient temperature and pressure (SATP) conditions (298.15 K; 1 bar), the molar mass of formaldehyde $M_{HCHO}$ and the stripping efficiency $\alpha$ are necessary to calculate the HCHO mixing ratio from the measured signal:

$$[\text{HCHO}] = \frac{(S - S_0)}{E_{Cal}} \frac{F_{liq}}{F_{gas}} \frac{V_{mol}}{\alpha M_{HCHO}} \tag{1}$$

The instrument sensitivity ($E_{Cal}$) is determined in daily calibration measurements using an internal, temperature-controlled

($T = 318$ K) HCHO permeation source, which provides a constant mass flow of HCHO ($\overset{*}{m}_{perm}$). The sensitivity can be calculated from the liquid flow rate $F_{liq}$ and the measured signal $S_{Cal}$ during calibration using Equation 1:

$$E_{Cal} = \frac{F_{liq} (S_{Cal} - S_0)}{\overset{*}{m}_{perm}} \tag{2}$$

Because the instrument can also measure liquid formaldehyde concentrations, the permeation source strength ($\overset{*}{m}_{perm}$) was regularly compared to a liquid formaldehyde standard solution ($\sim$ 0.1 mM HCHO) in 50 mM aqueous $H_2SO_4$ (the stripping

solution). Preparation of this standard required an intermediate dilution step ($\sim$ 10 mM HCHO in water) from a commercial standard solution of formaldehyde (37% HCHO w/w, Sigma-Aldrich). This stock solution was stored for multiple use in a refrigerator, in order to reduce polymerisation and evaporation (Golden and Valentini, 2014). Before each use, the concentration of the stock solution was determined titrimetrically: 10 mL of the stock solution and 30 mL of a freshly prepared 0.1 M aqueous sodium sulfite solution were automatically titrated to pH = 7 with a 1 N sulphuric acid solution by using a semi-

automatized titraton unit (716 DMS Titrino by Metrohm). The result was automatically calculated and used for the source strength calibration. The reproducibility of this procedure was 2 %. The uncertainty of the titration method is mainly due to the uncertainty of 3 % in the liquid flow measurements. Therefore, the combined accuracy of the Hantzsch method in this work is 8.5 % (Table 1).

## 2.3    Cavity ring-down spectroscopy (CRDS)

The second instrument for the detection of HCHO in this work uses Cavity Ring-Down Spectroscopy (CRDS) for the simultaneous detection and quantification of HCHO, $CH_4$ and $H_2O$ (Picarro, G2307) (Hoffnagle et al., 2017; Russel et al., 2020) that follows the same principles like other CRDS instrument from Picarro (Crosson, 2008). The gas concentration analyzer was placed in an air-conditioned container. A flow of 4 to 6 L min$^{-1}$ of air was sampled through a 4 m long 1/4 ” O.D. Teflon tube





from the chamber from which $0.28\ \mathrm{L\ min^{-1}}$ was taken by the CRDS instrument. The tip of the inlet in the chamber was close
to the inlet line of the Hantzsch instrument.

Cavity ring-down spectroscopy provides a very sensitive absorption measurement. The instrument consists of a cavity with
three triangularly arranged mirrors with high reflectivity. Light from a narrow-bandwidth laser is coupled into the cavity. Light
leaking through one of the mirrors is recorded by a photo-detector. The signal received from the photo-detector decays expo-
nentially after the laser has been switched off. The decay rate depends on the light loss in the cavity. This is partly due to the
presence of absorbers such as HCHO. Due to the high reflectivity of the mirrors an effective optical path length of several
kilometres is achieved. The wavelength of the laser is tuned within the range from $5625.5\ \mathrm{cm^{-1}}$ to $5626.5\ \mathrm{cm^{-1}}$ at a repetition
rate of 100 Hz, in order to simultaneously detect HCHO, $CH_4$ and $H_2O$ using specific rotational and vibrational absorption
lines. The overall accuracy of formaldehyde measurements is 10 % (Russel et al., 2020). The instrument automatically calcu-
lates concentrations of trace gases taking into account spectral overlap of the different absorbers and pressure and temperature
inside the cavity that are monitored (Barry et al., 2002; Hoffnagle et al., 2017).

Data are internally averaged to 1 Hz by the instrument and are averaged to 60 s time resolution for further analysis to
improve the precision of data. The instrumental precision and accuracy are reduced, if the humidity rapidly changes, because
of the overlapping absorptions lines of water and HCHO. Peak shapes also change with changes of temperature and pressure
and can impact the result of the peak fitting procedure (Picarro Inc. personal communication). Therefore, the cavity is pressure-
and temperature-stabilized (Picarro Inc.).

## 2.4 Differential optical absorption spectroscopy (DOAS)

A high-resolution laser differential optical absorption spectroscopy (DOAS) instrument provided absolute HCHO measure-
ments (Dorn et al., 1995; Schlosser et al., 2007, 2009). Light from a sub-picosecond pulsed, frequency-doubled dye laser
provides UV radiation around 308.04 nm with a bandwidth of 0.41 nm. The dye laser is synchronously pumped by a fre-
quency doubled model-locked Nd:YAG laser. The light passes through the central axis of the chamber in a White cell type
multi-reflection cell whose mirrors are installed at each end of the chamber in a distance of 20 m resulting in a total absorption
path length of $(2240 \pm 2)$ m. The light intensity transmitted through the chamber is spectrally analysed by a high resolution
Echelle grating spectrograph ($\Delta\lambda = 2.7\ \mathrm{pm}$, $\lambda/\Delta\lambda = 114\,000$, $f\ =\ 1.5\ \mathrm{m}$) and detected by a linear photodiode array detec-
tor.

The DOAS method relies on the separation of narrow absorption lines of specific absorbers from light attenuation that
does not vary much with wavelength. In the evaluation, the wavelength-dependent intensity of the transmitted light is fitted
to a polynomial to account for broadband extinctions and a superposition of differential absorption spectra from specific
absorbers. In order to calculate absorber concentrations, Lambert-Beer's law can be applied, but instead of using the total
absorption cross section, the differential absorption cross section ($\sigma_{diff}$) needs to be used. Similar to the analysis of the
transmitted spectrum, also the differential absorption cross section only contains the narrow absorption lines. The differential
absorption cross section for formaldehyde is $9.39\ \times\ 10^{-21}\ \mathrm{cm^2}$ at the maximum of the absorption in the wavelength region
used in the DOAS instrument here (308.1034 nm). The value of the absorption cross section was derived from the cross-



calibration between measurements of the DOAS and a Hantzsch instrument. The cross-calibration was achieved by comparing measurements in all chamber experiments in SAPHIR between 2011 and 2018, in which both instruments measured HCHO.

The slope of a regression analysis between both measurements were used to scale the differential absorption cross section that was applied in the evaluation procedure of the DOAS signal. It is worth noting that measurements used in the cross calibration are not part of comparison in this work. In addition, the Hantzsch instrument that delivered HCHO concentrations between 2011 and 2018 was different from the one used in this work. Therefore, DOAS measurements can be regarded as independent from Hantzsch measurements in the comparison here.

High selectivity is achieved by the high resolution of the measured spectrum that allows separation of overlapping narrow-band spectral structures. The DOAS system used in this work can simultaneously detect hydroxyl radicals (OH), HCHO, $SO_2$ and naphthalene. OH radicals are measured with a very high spectral resolution in the same spectral range as for HCHO, making it an ideal component for a spectral reference. This OH reference spectrum can be monitored and compared to the actual OH measurements periodically for the evaluation of the stability of the spectral resolving power and the repeatability of

the spectral wavelength scanning mechanism (Dorn et al., 1995; Hausmann et al., 1997). Both are important for maintaining repeatable and accurate measurements recorded by the Echelle spectrometer.

Formaldehyde measurements by DOAS have a time resolution of 135 s with a stated 1 $\sigma$ precision of 0.5 ppbv (Brauers et al., 2007). The detection limit is mainly limited by the residual structures in the spectra. The combined accuracy of 6 % is basically derived from the uncertainty of the absorption cross section, which is a function of ambient pressure and temperature

(Cantrell et al., 1990) and additionally depends on the instrumental response function of the detection system.

## 3 Results and discussion

### 3.1 Stability of the instrument zero and sensitivity of the Hantzsch instrument

The instrumental zero of the Hantzsch instrument exhibits fluctuations and drifts. This is likely caused by temperature changes of parts of the instrument that are not temperature-controlled. Therefore, the Hantzsch instrument determines the instrument

zero from regular, automatized measurements, in which formaldehyde is removed from the sampled air. HCHO concentrations automatically provided by the instrument are calculated according to Equation 1. By default the last zero measurement for $S_0$ is used until the next zero measurement is done. As a consequence, the time series of HCHO concentrations can show artificial jumps after a new zero reading has been taken, if the zero value has changed (Fig. 1).

In order to smooth the fluctuations caused by changes in the zero signal, data in this work were reprocessed by applying a

linear interpolation of the zero measurements before and after the actual HCHO measurement. In addition, the time interval between two zero measurements was reduced from 4 hours to 2 hours after the significant changes in the instrument zero had been noticed. Figure 1 shows that changes of the instrument zero can be as high as 0.2 mV, which is equivalent to HCHO mixing ratios of up to 1.2 ppbv. The exact value depends on the current sensitivity of the instrument. By applying more frequent zero measurements and interpolating between zero measurements, the uncertainty in the HCHO measurements could



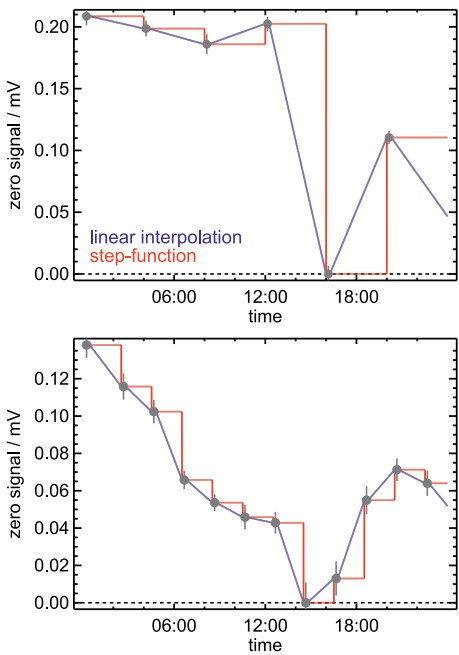

**Figure 1.** Plot of instrument zeros for two days of measurements, if either only the last zero measurement is used (step function) or a linear interpolation between two zero measurements. Error bars represent 1 $\sigma$ errors of the zero measurements. The change of the raw signal of 0.2 mV corresponds to a HCHO mixing ratio of 1.2 ppbv.

be significantly reduced by a at least a factor of 10, so that the accuracy of measurements due to the uncertainty in the zero is well below 100 pptv.

The Hantzsch instrument was in operation at the SAPHIR chamber for nearly half a year. During experiment episodes, calibration measurements were done once a day. This allows analysing the stability of the instrument's sensitivity. Figure 2 shows the deviation from the mean sensitivity for 116 calibration measurements. The mean value of the sensitivity of the

instrument was 75 L mV µg$^{-1}$. The 1 $\sigma$ reproducibility of the calibration measurements was 5 % and has an accuracy of 8.5 % (Table 1). The record of calibration measurements indicates a good long-term stability of the instrument's sensitivity. Nevertheless, day-to-day changes are most likely due to real changes of the sensitivity, because the sensitivity is expected to decrease with ageing of the tubing. This is also clearly seen in the increase of the sensitivity after each exchange of the tubing (Fig. 2). Therefore, the calibration value that was measured close to the time of the actual measurement was applied for the

evaluation of measurements in this work.

## 3.2    Water vapour dependent offset in HCHO measurements by Picarro CRDS

Formaldehyde measurements by the CRDS instrument appeared to have a bias. An offset was noticed in measurements in synthetic air in the clean chamber, when no formaldehyde was present (Fig. 3). This offset is variable with time and depends





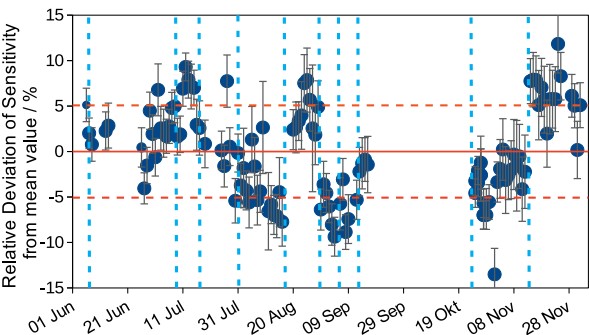

**Figure 2.** The Hantzsch instrument was calibrated on 116 days over a period from June to December 2020. Each point shows the relative deviation of the mean measured sensitivity. Error bars are standard deviations from the mean value of all data points acquired during one calibration measurement. Dashed red lines give the standard deviation of the sensitivity averaged over the entire period (5 %). Dashed vertical lines indicate times, when there was work done on the instrument such as exchanging of the tubes.

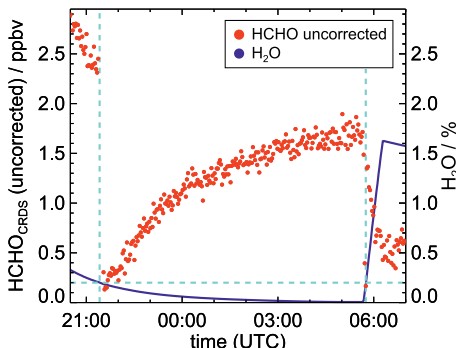

**Figure 3.** Time series of HCHO measurements by the CRDS instrument without offset correction with varying water mixing ratios in the chamber. Measurements by the Hantzsch instrument (not shown) indicated that mixing ratios were much smaller than the Picarro measurements with values that were continuously decreasing from 200 pptv to zero until midnight due to the flushing of the chamber with zero air. They remained close to zero also during the humidification starting at 05:45 in morning. In contrast, uncorrected CRDS measurements showed fast changes, when the water vapour mixing ratio became lower or higher than. 0.2 %. They also showed slow changes that (anti-) correlate with the water vapour mixing ratios. Before humidification an offset value of 1.7 ppbv is measured in dry air.





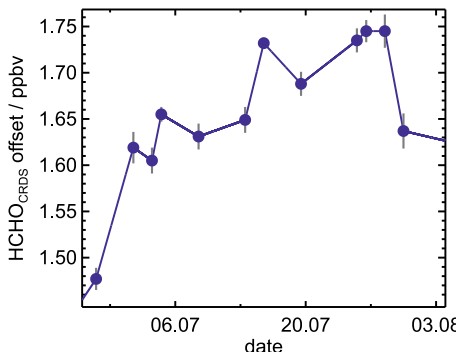

**Figure 4.** Measured offsets of the HCHO measurements by the CRDS instrument in dry air in the period from 26 June to 03 August 2019. Values were determined on 12 days by sampling air without HCHO and extrapolating the linear water vapour dependent bias of the instrument to dry conditions.

on the presence of water vapour. As seen in Fig. 3, a significant offset is also measured by the instrument in clean dry air.
The offset in dry air shows relatively small changes of typically less than $0.1\,\mathrm{ppbv}$ from day to day. In addition to the offset
in dry air, changes in humidity causes additional variations of the instrument offset of up to $2\,\mathrm{ppbv}$. The exact value scales
linearly with the water vapour mixing ratio. The offset in dry air can be derived by linearly extrapolating measurements without
formaldehyde to dry conditions. This could be done for experiments here, when the clean air in the chamber was humidified.
Figure. 4 shows the series the resulting offset values in dry air for all experiments. During this time the bias in the measurements
changed significantly over time with values between 1.3 and 1.75 $\mathrm{ppbv}$. All measurements in this work were corrected for this
bias by using the same value for dry conditions on each day. If no measurements without HCHO were available, a linearly
interpolated value from experiments on days before and after was taken. The variability of the bias is within the range of the
typical zero drift of $0.3\,\mathrm{ppbv}$ and the absolute value of the offset is in the range of the specified zero drift of $1.5\,\mathrm{ppbv}$ (Picarro
Inc.).

Figure 5 shows the water vapour dependence of the bias in the CRDS measurements that adds to the offset in dry air. In
order to make data of different days comparable, all measurements were corrected for the variability of the day-specific offset
in dry air. Again, data from the experiments between 26 June to 03 August 2019 in the SAPHIR chamber were taken, when no
HCHO was present. Measurements at water vapour mixing ratios lower than 0.2 % separate from the measurements at water
vapour mixing ratios higher than 0.2 %. Apparently, the intercepts for dry conditions are the same for both groups of data on a
particular day. Both subsets of data exhibit a bias that decreases linearly with increasing water vapour mixing ratio. However,
slopes of the linear relationship for lower water vapour mixing ratios are approximately a factor of 15 higher compared to
that for higher water vapour mixing ratios with a value of $(-11.20 \pm 1.60)\,\mathrm{ppbv}\,\%^{-1}$ for water vapour mixing ratios lower
than 0.2 % and a value of $(-0.72 \pm 0.08)\,\mathrm{ppbv}\,\%^{-1}$ for water vapour mixing ratios higher than 0.2 %. The water vapour
dependence is similar for all data. Therefore slopes were determined from linear fits using all data. These values combined
with the day-specific intercept were used to correct all CRDS HCHO data in this work.




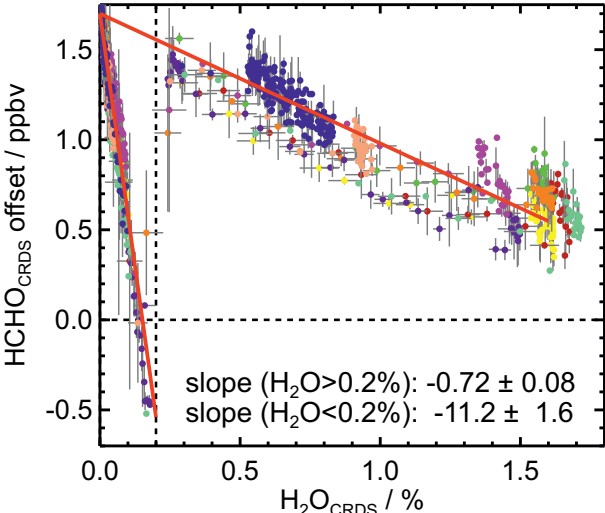

**Figure 5.** Offsets in the CRDS measurements with changing water vapour mixing ratio after removing the variability of the offset for dry conditions (Fig. 4). Data from 11 experiments are included indicated by different colors (number of data points: 1040 for $H_2O<0.2\%$, 541 for $H_2O>0.2\%$). Red lines show the results of a linear regression analysis ($R^2 = 0.92$ for $H_2O<0.2\%$, $R^2 = 0.83$ for $H_2O>0.2\%$). Gray error bars represent $1\,\sigma$ errors of measurements.

The observed changes in the offset is caused by the data processing algorithm in the instrument, which takes into account the spectral overlap of water and formaldehyde infra-red absorption lines (Picarro Inc. personal communication). A very strong water line interferes with the formaldehyde absorption. Above 0.2 % water mixing ratio, the water vapour absorption line is strong enough, so that its contribution can be accounted for in a fit of absorption lines in which amplitudes and line widths are free parameters. In contrast, below 0.2 % the signal-to-noise is too poor for an independent fit of the line width. Therefore, a fixed value for the line width which was derived from a spectrum that was acquired at very low water concentration is used. This procedure has been improved in new versions of the instrumental software (version 1.6.015 implemented in the instrument used here) (Picarro Inc. personal communication). Therefore, the correction described here might not be applicable for all Picarrao HCHO instruments.

## 3.3 Precision of measurements

In order to analyze the precision of measurements, the Allan deviation was calculated from measurements in air that did not contain formaldehyde (Fig. 6). The DOAS instrument did not provide a sufficiently high number of data points to perform an Allan deviation analysis. Corrections of data from the Hantzsch and CRDS instruments were applied as described above.

The Allan deviations for the CRDS and Hantzsch measurements result in $1\,\sigma$ precisions of 0.08 and 0.014 ppbv, respectively, at an averaging time of 2 min. The Hantzsch instrument provides an overall better signal-to-noise ratio and therefore a lower limit of detection compared to the CRDS instrument. Although the frequency with which data is provided by the





Hantzsch instrument can be in the range of seconds, the slow transport of the solution inside the instrument results in a time resolution of 90 s as demonstrated by the constant Allan deviation for averaging times of less than 90 s and specified by the manufacturer. The Allan deviation does also not decrease as expected from White noise for longer averaging times. This indi-

cates that measurements are impacted by time-dependent, systematic errors. These could be for example drifts of the baseline that are not fully taken into account as seen in Fig. 4, but also short-term variability in the instrument sensitivity. In contrast, CRDS data follow White noise up to an averaging time of 2 hours. For example, the Allan deviation is approximately 50 pptv for an integration time of 5 minutes consistent with the typical precision that is specified for the instrument (Picarro Inc.).

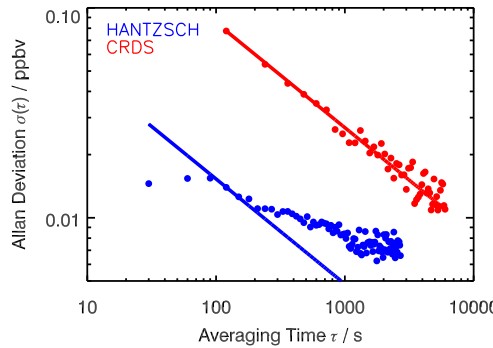

**Figure 6.** Allan deviation (1 $\sigma$ precision) calculated from zero measurements in the clean, dark chamber. Straight lines represent White noise $(\sigma(\tau)^2 \propto \tau^{-1})$.

## 3.4   Comparison of measurements

From June 2019 to December 2019 numerous experiments were performed in the SAPHIR chamber. In photochemical experiments studying the photo-degradation of organic compounds, the chamber was operated with synthetic air, in which trace gases were injected. During the JULIAC campaign, the chamber was filled with ambient air. The diversity of experiments allowed for measuring HCHO for a wide range of conditions regarding temperature, relative humidity, ozone, nitrogen oxides, and methane concentrations (Table 2).

Regular flushing of the chamber with pure synthetic air provided a solid instrumental zero for most instruments and specific oxidation experiments provided high levels of nitric oxides and ozone up to peak values of 60 ppbv and 600 ppbv, respectively. For instance, formaldehyde was monitored during the photo-oxidation of cyclic monoterpenes such as limonene, carene, $\alpha$-pinene, $\beta$-pinene, isoprene and alkenes. In these batch experiments levels of nitrogen oxides and ozone were variable, in order to influence chemical oxidation and degradation process of aliphatic hydrocarbons and therefore the formation of formaldehyde.

The comparison of all formaldehyde measurements by the three instruments described above are shown as time series in Fig. 7 and as correlation plot in Fig. 8. Hantzsch measurements were interrupted for regular instrumental maintenance or calibration. Each exchange of peristaltic tubes or a power off for maintenance required a longer time until the measurements became again





**Table 2.** Range of conditions inside the SAPHIR chamber in the time between June 2019 and December 2019.

|  | Min | Max | Median |
|---|---|---|---|
| formaldehyde / ppbv | 0 | 11 | 0.5 |
| abs. water / % | 0 | 2.6 | 0.7 |
| rel. humidity / % | 0 | 85 | 26.6 |
| temperature / $^\circ$C | −1 | 42 | 17.3 |
| nitrogen oxides / ppbv | 0 | 60 | 0.9 |
| ozone / ppbv | 0 | 170 | 12.1 |
| methane / ppmv | 0 | 17 | 1.9 |
| hydroxyl radicals / $10^7 \mathrm{cm}^{-3}$ | 0 | 2 | 0.4 |

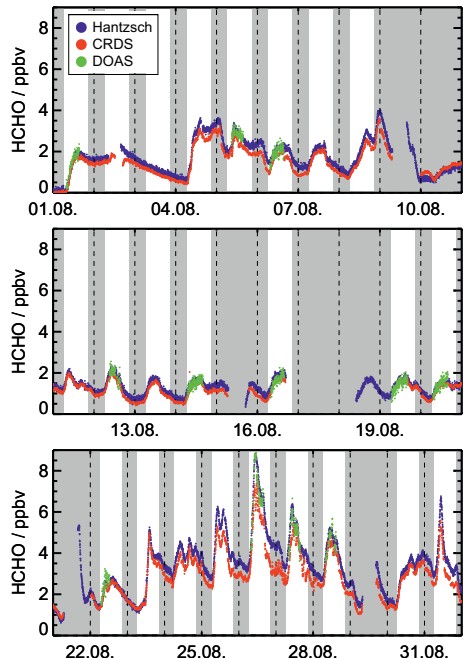

**Figure 7.** Formaldehyde measurements from Hantzsch, CRDS and DOAS instrument in experiments in the SAPHIR chamber during August 2019, when the chamber was continuously flushed with ambient air (JULIAC campaign). Coloured grey areas indicate nighttime or when the chamber roof was kept closed during daytime and dashed vertical lines indicate midnight.

stable. Also CRDS and DOAS needed maintenance times explaining regular gaps in the measurements. Measurements by the Hantzsch and Picarro instruments are corrected for variable offsets as discussed in the previous sections.

Figure 7 shows an exemplary time series from the JULIAC campaign in August 2019. Formaldehyde varied from 0 to 8 ppbv. During the last week of August ambient temperature and ozone was high with up to 40 $^\circ$C and 100 ppbv, respectively.





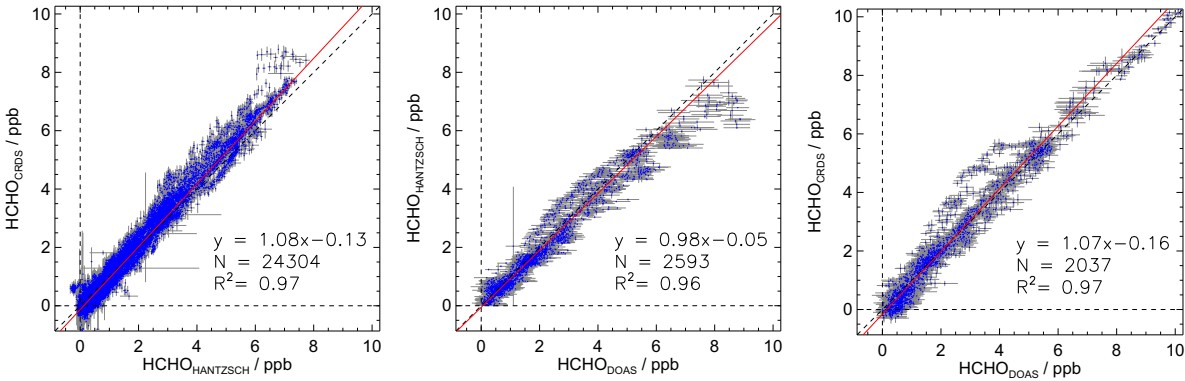

**Figure 8.** Correlation plots of measurements from Hantzsch, CRDS and DOAS instruments during experiments in the SAPHIR chamber. Data were acquired between June to December 2019 (time resolution 180 s). Red lines show results from the regression analysis. Error bars represent 1 $\sigma$ errors of measurements.

Production of formaldehyde is likely enhanced by a high oxidation rate of organic compounds. Time series of measurements of all three instruments exhibit the same behaviour and show an overall good agreement in the measured concentrations. During several periods like between 23 August and 31 August, systematic deviations within the range of a few hundred pptv are observed. No specific reason could be identified that would explain these deviations. Most likely, either unaccounted small changes in the calibration or in the offset values caused these differences.

Figure 8 shows the correlation and regression analysis (time resolution: 180 s, weighted fit with errors in both coordinates) of the entire data set between 01 June and 03 December 2019. Data from CRDS and Hantzsch instruments contain more than 24,000 data points and show very good agreement ($R^2 > 0.96$). This confirms the applicability of the two different methods for atmospheric formaldehyde measurements. Measurements by the CRDS instrument are 8 % higher compared to the measurements by the Hantzsch instrument. The intercept of the regression line is 0.13 ppbv. The comparison of Hantzsch and DOAS measurements (2,593 data points) also results in a high linear correlation coefficient of $R^2 = 0.96$. DOAS measurements are on average 2 % higher than those of the Hantzsch instrument. The linear regression between CRDS and DOAS measurements (2,037 data points) results in a slope of 1.07 and an intercept of –0.16 ppbv with a linear correlation coefficient of $R^2 = 0.97$. The statistical errors of each of the fit parameters of the linear regressions are lower than 0.01 due to the small errors of single data points. All deviations of the slopes from unity are within the combined accuracies of instruments (Table 1). No systematic deviations in the measurements are identified and no corrections other than those described above need to be applied.

Instruments for the detection of formaldehyde have been compared before. In 2005, Hak et al. reported several comparisons of measurements by a Hantzsch and a broadband-DOAS instrument. Comparisons showed various results ranging from good agreement within the uncertainties of measurements to significant over- or underestimations. However, no consistent explanation for these variable results could be found. In 2008, Wisthaler et al. (2008) reported the comparison of formaldehyde measurements from five instruments based on four different measurement techniques. Measurements were performed in the





SAPHIR chamber during 5 day-long experiments in January 2005. Measurement techniques included Hantzsch, broadband-DOAS, HPCL and PTR-MS. An overall good agreement of all measurements were found with one exception. Significantly
lower formaldehyde concentrations were observed in the Hantzsch and HPLC instruments, if ozone was present (up to a mixing ratio of 44 ppbv) (Wisthaler et al., 2008). Interferences from ozone, water or methane in the HCHO measurements by the Hantzsch instrument were also found in other comparisons (Grossmann, 2003; Warneke et al., 2011). The manufacturer of the Hantzsch instrument states that ozone can interfere in HCHO measurements. Artificial signals equivalent to 1 ppbv HCHO are possible for 800 ppbv $O_3$ (Aero-Laser GmbH). For conditions experienced in experiments in this work, however, such an
interferences would be negligible. A recent review about spectrophotometric methods for formaldehyde detection reported high sensitivity of the Hantzsch derivatisation reaction with negligible chemical interferences and only minimal sensitivity to other aldehydes and ketones (Hladová et al., 2019). This agrees with findings here, which do not hint to systematic cross-sensitivities for atmospheric conditions.

## 4   Conclusions

More than 100 days of measurements were performed by instruments detecting formaldehyde in photochemical experiments in the SAPHIR chamber in 2019. Two commercial instruments (CRDS by Picrarro and Hantzsch by AeroLaser) and one custom-built, high-resolution DOAS instrument measured together. Experiments included 56 day-long experiments, when the chamber was continuously flowed with ambient air (residence time approximately one hour). Measurements were performed during day and night. In addition, HCHO was measured in experiments investigating the photo-oxidation of specific organic compounds.
Physical and chemical conditions were variable. Formaldehyde mixing ratios were similar as found in field campaigns with values between 0 and 8 ppbv (Inomata et al., 2008; Leuchner et al., 2016). The comparison of measurements by the three instruments included more than 24,000 data points for Hantzsch and CRDS instruments and 2,600 data points for the DOAS instrument.

    The analysis of measurements revealed that instrumental zeros of the Hantzsch and CRDS need to be carefully treated.
The background signal of the Hantzsch instrument can significantly change within a few hours. Therefore, regular zero measurements at least every 2 hours are required to avoid systematic errors that can be as high as several 100 pptv. In addition, post-processing of data can reduce systematic errors. Zero values for each individual data point can then be calculated from a linear interpolating of zero measurements. In contrast, the actual output values of the instrument takes only the last zero measurement into account, so that systematic errors can occur, if the instrument zero changes between two consecutive zero
measurements.

    The CRDS instrument also shows an instrumental zero that varies within days. Changes in the range of 100 pptv per day were observed in these experiments. This drift is within the specifications of the instrument, but could be relevant for measurements in environments with low formaldehyde mixing ratios. In addition, the background signal exhibits a significant water dependence. The value decreases linearly with increasing water vapour mixing ratio, but the slope of the linear relationship
changes drastically for water mixing ratios higher than 0.2 %. The water dependence is a factor of 15 lower compared to the de-

pendence at water mixing ratios lower than 0.2 %. The values of the slopes for both ranges of water vapour mixing ratio appear to be similar for all measurements in this work. Daily measurement of the instrumental zero and regular characterization of its water vapour dependence is required to avoid systematic errors. In addition, the manufacturer stated that the mathematical process of peak integration and internal data evaluation was changed in an updated version of the instrument (software version

1.6.015 is implemented in the instrument used in this work). Therefore, the water vapour dependence may be different in other Picarro HCHO analyzers.

    The large data set of formaldehyde data measured by the Hantzsch, CRDS and DOAS instruments shows good agreement ($R^2 \geq 0.96$) between all methods. Maximum deviations of the slopes of the regression analysis from unity are 0.08. The combination of wet-chemical conversion with sensitive fluorescence detection provided a very good limit of detection in the

experiments in this work, which is consistent with the instrumental specification and results found in other studies (Hak et al., 2005; Salmon et al., 2008). The limit of detection of the CRDS instrument is $200\,\mathrm{pptv}$ ($3\,\sigma$) at a time resolution of $180\,\mathrm{s}$. This is sufficient for the application of the instrument in many cases. The limit of detection of the CRDS instrument can be further lowered, if data is averaged for a longer time.

    Results here show that the three instruments provide reliable and accurate formaldehyde measurements in ambient air, if the

instrumental zeros of the CRDS and Hantzsch instruments are adequately taken into account. Among the two commercially available instruments the Hantzsch instrument provides the higher sensitivity, which is beneficial for measurements in environments with low HCHO concentrations. On the other hand, the CRDS instrument has the advantage of being small-sized. In addition, largely unattended operation is possible, because regular calibrations or maintenance work are not required. This is ideal for long term measurements in environments where the slightly higher detection limit does not pose a problem.

*Data availability.* Data will be made available from the Jülich Data database and can be referenced with a DOI latest at the time of final publication.

*Author contributions.* MG was responsible for Hantzsch measurements and analysed the data. He wrote the paper together with HF. AH designed JULIAC campaign and organized the campaign together with HF and FH. FR was responsible for the CRDS and HPD for DOAS measurements. All co-authors contributed to the discussion of the manuscript.

*Competing interests.* The authors declare that they have no conflict of interest.

*Acknowledgements.* This project has received funding from the European Research Council (ERC) under the European Union's Horizon 2020 research and innovation program (grant agreement no. 681529).).





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
