# Peer review of "Comparison of formaldehyde measurements by Hantzsch, CRDS and DOAS in the SAPHIR chamber"

_Atmospheric Measurement Techniques, 2021_

## Referee Comment (RC1)

Review of "Comparison of formaldehyde measurements by Hantzsch, CRDS and DOAS in the SAPHIR chamber," Glowania et al., AMT (2021)

This manuscript describes a comparison of three measurements of atmospheric formaldehyde (HCHO). The instrument were run simultaneously during a series of experiments in the SAPHIR chamber. Corrections are developed for the Hantzsch and CRDS methods to account for drifting baselines and water vapor artifacts. After these corrections, all three measurements agree to within their stated uncertainties. This paper is appropriate for publication in AMT. Suggested revisions are mostly minor.

**Specific Comments**

L112: There are also applications in the 10s and 100's of ppt range, where there are different challenges.

L186 – 189: is this proprietary information? This "rod-shaped material" sounds like Hopcalite or a similar catalyst. Would this material be recommended as a zero-method for the CRDS?

L254: This implies that the DOAS is not "calibration-free" as stated in the abstract. It was calibrated against a Hantzch. Rather it doesn't need to be continuously calibrated. Recommend clarifying language.

L263: Does this accuracy also propagate the uncertainty in the Hantzsch method used to calibrate the differential cross section? Also, does this match current recommendations by, e.g., the JPL handbook? HCHO UV cross sections changed by 8% between JPL 2011 and 2016, and these are based on Meller and Moortgat (2000).

Figure 1: A plot comparing the mixing ratios calculated using either method might be more informative.

L283: Is the timing of calibration measurements random?

L287: While this empirical correction method may work for this particular set of experiments, it does not provide a recipe that is easily generalized for other CRDS users. How rapidly does this offset change over a day? Is there a dependence on ambient or instrument temperatures? Such questions could be answered with lab experiments under controlled conditions (sampling the same concentration and systematically varying H2O or temperature), and it is somewhat surprising that such experiments were not done in this case to provide robust empirical correction algorithms.

L335: would these "time-dependent systematic errors" be less evident if the test were done at higher HCHO (well above the instrument detection limit)?

General comment: do the authors recommend implementing an automated zero system (background determination) for the CRDS? If so, how frequent should zeros be to achieve good accuracy? Should the method of zeroing retain ambient water vapor concentrations (e.g. using a catalyst) or can one get away with drierite or zero air? This is considering uses like routine monitoring for air quality.

**Technical Comments**

L63: maybe worth noting that this detection limit is with a data frequency of 1 Hz. Also, maybe worth also citing Cazorla et al. (2015, AMT).

L77: delete "the sensitivity is not satisfying and"

---

## Referee Comment (RC2)

Review of Glowania et al., AMT (2021):

This manuscript presents a comparison of HCHO measurements taken at the SAPHIR chamber over several experiments. The analysis focuses on the performance of two commercially available HCHO instruments using Hantzsch (Aero-Laser GmbH, AL4021) and CRDS (Picarro, G2307) techniques and a DOAS instrument. The authors conclude that zero-corrected measurements are in agreement within stated uncertainties for all measurement techniques. The subject matter is appropriate for AMT. Specific comments are enumerated below.

Specific comments:

Line 64-66: The LIF technique does not require a fiber laser (see Hottle et al., 2009), though a fiber laser is often used. Please include Cazorla et al., 2015 as a reference.

Table 1. This reads as an assessment of the measurement methodologies (Hantsch/CRDS/DOAS), though the figures given refer only to the specific instruments used here. I suggest titling each column with the instrument model.

Line 186. Please provide the name of the reagent used for HCHO scrubbing.

Line 206. It is unclear to me how the reproducibility of 2% on the titration and a 3% uncertainty in flow measurements adds together to get an 8.5% measurement accuracy. Please clarify.

Line 214. Please quantify "close".

Line 223. It is unclear where the stated accuracy of 10% comes from. It is not derived or demonstrated in this study. The referenced paper (Russel et al., 2020) discusses the Picarro instrument uncertainty in the supplement. It states "The CRDS from Picarro is factory calibrated and has a precision of 1.2 ppb + 0.1% of the reading for HCHO readings, with no dependence on humidity levels […] For the determination of absolute concentrations of HCHO, for instance during chamber testing, the accuracy is ± 10%". Glowania et al. show a humidity dependence and gives a higher precision. It seems the two papers are not in agreement, so it does not seem that the accuracy in the reference given can be applied here.

Line 253. The DOAS method relies on cross-calibration with a Hantzsch. While two different Hantzsch instruments are used, is it possible that any systematic offsets related to the Hantzsch methodology could influence DOAS observations? I suggest showing the data used in the regression analysis discussed in lines 248-252.

Line 263-265. Again, it is unclear how the 6% accuracy is calculated. More details are needed. Does it take into account the uncertainty in the absorption cross section from the regression described in lines 248-252?

Figure 1. I suggest showing HCHO concentrations using each method in panel 2, rather than a second day showing the same thing as panel 1.

Line 291. If the Picarro specifies an anticipated zero drift of 1.5 ppb, I think the offset should not be classified as a "bias", but rather a zero-point that needs to be taken into account. The offset of the Hantzsch instrument is not called a "bias".

Line 366: Is the intercept -0.13 ppb?

Lines 406-416. It is unclear how the main points of this paper can be applied to future deployments of the CRDS method. How often does a HCHO zero need to be taken with the CRDS instrument if ambient $H_2O$ varies? Could the HCHO zeroing system used for the Hantzsch be applied to the Picarro? Can the authors reassess their observations using the new software?

Throughout. The authors switch between $1\sigma$ and $3\sigma$ LODs. A consistent reporting methodology would improve the readability.

---

## Referee Comment (RC3)

Review of Glowania et al., AMT (2021):

The manuscript presents data from three different HCHO measurement techniques that were collected over a months-long period sampling environmental chamber experiments. Since this appears to be the first intercomparison to evaluate the commercially available Picarro instrument and also provides valuable advice on Aero Laser instrument operation, it is a service to the community and worth publishing. Considering the magnitude and uncertain nature (e.g. which version of Picarro fitting software?) of variable offsets, this reviewer is left to conclude that there is still work to do before these commercial HCHO instruments can consistently provide accurate measurements at the 0-3 ppbv level. A few comments are provided below.

Line 65: LIF for HCHO does not require a custom fiber laser, e.g., St. Clair et al., 2019. https://doi.org/10.5194/amt-12-4581-2019

Line 83: 'Comparisons' shouldn't be capitalized.

Line 211: The Russell et al., 2020 citation uses the Picarro instrument, but provides no understanding of the instrument itself and adds no value to this manuscript. Perhaps it can be considered an instrument intercomparison, but that seems a real stretch.  Your work here is considerably better on that topic.

Line 111: *"Reports of instrument comparisons concluded that the measurement of formaldehyde remains challenging specifically for atmospheric concentrations in the low ppbv range."*
This may be true for commercially available instruments, but is not true for research-grade instruments. You should qualify this statement.

Line 253: *"Therefore, DOAS measurements can be regarded as independent from Hantzsch measurements in the comparison here."*
I strongly disagree with this statement. The DOAS data is produced using an empirical calibration where a Hantzsch instrument was the standard. They each may provide some unique information for data evaluation, but they should not be considered independent measurements. On that point, I don't understand how the DOAS measurement can have a higher accuracy (6%) than the technique used to calibrate it (Hantzsch, 8.5%).

Section 3.2: How do you know that the size of the water-dependent offset determined at HCHO=0 is the same size as the water-dependent offset in the presence of HCHO? Since this is a spectroscopic interference, it seems possible that the fitting error caused by water will be dependent on the magnitude of the HCHO signal as well. A zeroing approach that removes HCHO but preserves humidity would be one solution. Is the 1.5 ppbv zero drift specified by Picarro due to this water interference or due to other factors?

Figure 7 is rather small for the amount of data it contains. Please consider making it bigger.

---

## Author Response (AR1)

Response to the comments of referee #3:

We thank the reviewer for the helpful comments.

Comment: L112: There are also applications in the 10s and 100's of ppt range, where there are different challenges.

Response: We added in L112: "...for atmospheric concentrations in the low and sub- ppbv range."

Comment: L186-189: Is this proprietary information? This "rod-shaped material" sounds like Hopcalite or a similar catalyst. Would this material be recommended as a zero-method for the CRDS?

Response: Unfortunately, the manufacturer did not provide which exact material is used, though we asked for this information. Therefore, we cannot give more information than currently done. To make this point clear, we added: "However, the exact scavenging material is not specified by the manufacturer."

Comment: L254: This implies that the DOAS is not "calibration-free" as stated in the abstract. It was calibrated against a Hantzsch. Rather it does not need to be continuously calibrated. Recommend clarifying language.

Response: We replaced the "cross-calibration" in line 247/248 by "comparison" and cancelled "calibration-free" in the abstract.

Comment: L263: Does this accuracy also propagate the uncertainty in the Hantzsch method used to calibrate the differential cross section? Also, does this match current recommendations by, e.g., the JPL handbook? HCHO UV cross sections changed by 8% between JPL 2011 and 2016, and these are based on Meller and Moortgat (2000).

Response: The rotational-vibrational absorption lines of HCHO around 308 nm are very narrow (in the low picometer range) and consequently the experimentally observes cross section is strongly dependent on the spectral resolution of the instrument. Meller and Moortgat used a resolution of 0.025 nm while our DOAS instrument has a resolution of 0.0025 nm and therefore the cross sections are different. We changed the text L263: "The accuracy of the DOAS formaldehyde data is estimated to be 7%. It is mainly determined by the accuracy of the calibration procedure of the former Hantzsch instrument which was used for the comparison with the DOAS. It also takes into account the uncertainty in the absorption cross section from the regression between the DOAS and Hantzsch data which was 0.16%. The derived high-resolution absorption cross section is valid for the given spectral resolution of 0.0027 nm. Under ambient conditions the pressure and temperature dependence of the cross section is very small (Cantrell et al., 1990) and does not affect the accuracy of the DOAS measurements."

Comment: Figure 1: A plot comparing the mixing ratios calculated using either method might be more informative.

Response: In our opinion, plotting calculated mixing ratios of a zero signal does not illustrate exactly what we intend to show. The point we would like to make is that there is a changing zero signal that needs to be adequately monitored. Plotting mixing ratio would require subtracting the changing zero signal from the measurements and the information, how much the zero signal changes over time is lost. Therefore, we prefer keeping the plot as it is. The information how the signal converts to an equivalent HCHO concentration is given in the caption.

Comment: L283: Is the timing of calibration measurements random?

Response: Yes, calibration was performed, when there was no experiment performed in the chamber. Breaks were not done on a regular basis but were done according to the schedule of experiments.

Comment: L287: While this empirical correction method may work for this particular set of experiments, it does not provide a recipe that is easily generalized for other CRDS users. How rapidly does this offset change over a day? Is there a dependence on ambient or instrument temperatures? Such questions could be answered with lab experiments under controlled conditions (sampling the same concentration and systematically varying

H2O or temperature), and it is somewhat surprising that such experiments were not done in this case to provide robust empirical correction algorithms.

Response: This work focusses on the results of the application of the instrument in ambient conditions. In our opinion the type of experiments does not play a role for the observed effects. The instrument was placed in an air-conditioned sea container. This type of deployment is very typical, so that the performance of the instrument can be expected to be similar in many cases. In addition, the ring-down cavity is pressure and temperature stabilized, so that the variability of the offset cannot be explained by effects from the specific deployment. Recommendations for the required characterization of the instrument are given in the text. The instrument's offset needs to be measured at least once a day and the dependence on water needs to be characterized on a regular basis. From the drifts of parameters we observe, we do not expect that there a correction with fixed numbers can be provided, but parameters need be determined for each individual instrument and deployment.

Comment: L335: would these "time-dependent systematic errors" be less evident if the test were done at higher HCHO (well above the instrument detection limit)?

Response: The Allan deviation is calculated for baseline measurements without HCHO. The relative impact of the baseline noise would be small for high HCHO concentrations, but the purpose of the Allan deviation is to estimate the limit of detection of instruments.

Comment: General comment: do the authors recommend implementing an automated zero system (background determination) for the CRDS? If so, how frequent should zeros be to achieve good accuracy? Should the method of zeroing retain ambient water vapor concentrations (e.g. using a catalyst) or can one get away with drierite or zero air? This is considering uses like routine monitoring for air quality.

Response: The need for an automated zero system depends on the accuracy that the user needs. Our observation demonstrate drifts are within the specification of the manufacturer, but that there is need for regular zero measurements, if the user wants to achieve a higher accuracy. From what we see, zeroing once a day would be sufficient and characterization of the the humidity dependence on a regular basis. Our dataset does not allow to specify how often this needs to be done. We added L425: "Observations in this work suggest that zero measurements should be done once a day and that the water dependence of the zero point of the CRDS instruments does likely not significantly change at least for a month-long deployment."

Comment: L63: maybe worth noting that this detection limit is with a data frequency of 1 Hz. Also, maybe worth also citing Cazorla et al. (2015, AMT).

Response: Done.

Comment: L77: delete "the sensitivity is not satisfying and"

Response: Done.

Response to the comments of referee #2:

We thank the reviewer for the helpful comments.

Comment: Line 64-66: The LIF technique does not require a fiber laser (see Hottle et al., 2009), though a fiber laser is often used. Please include Cazorla et al., 2015 as a reference.

Response: We added the citation and changed the text: "LIF instruments also custom-built and instruments often make use of a fibre laser for the excitation."

Comment: Table 1. This reads as an assessment of the measurement methodologies (Hantsch/CRDS/DOAS), though the figures given refer only to the specific instruments used here. I suggest titling each column with the instrument model.

Response: We added the instrument model as an additional row.

Comment: Line 186. Please provide the name of the reagent used for HCHO scrubbing.

Response: Unfortunately, the manufacturer did not provide which exact material is used, though we asked for this information. Therefore, we cannot give more information than currently done. To make this point clear, we added: "However, the exact scavenging material is not specified by the manufacturer."

Comment: Line 206. It is unclear to me how the reproducibility of 2% on the titration and a 3% uncertainty in flow measurements adds together to get an 8.5% measurement accuracy. Please clarify.

Response: In order to explain, how the accuracy is calculated we changed the text L206: „Error propagation results in a total uncertainty of the HCHO concentration of 3.6,% that adds to the reproducibility of the calibration procedure of 5% (Section 3.1)." We changed also the text in L285: "The 1 sigma reproducibility of the calibration measurements was 5% and has been included in the HCHO measurement accuracy of 8.6% (Table 1)."

Comment: Line 214. Please quantify "close".

Response: We changed the text to "approximately 0.5m away from".

Comment: Line 223. It is unclear where the stated accuracy of 10% comes from. It is not derived or demonstrated in this study. The referenced paper (Russell et al., 2020) discusses the Picarro instrument uncertainty in the supplement. It states "The CRDS from Picarro is factory calibrated and has a precision of 1.2ppb + 0.1% of the reading for HCHO readings, with no dependence on humidity levels [...]. For the determination of absolute concentrations of HCHO, for instance during chamber testing, the accuracy is $pm10$%". Glowania et al. show a humidity dependence and gives a higher precision. It seems the two papers are not in agreement, so it does not seem that the accuracy in the reference given can be applied here.

Response: The number is taken from the datasheet provided by Picarro (https://www.picarro.com/sites/default/files/Picarro_G2307%20Datasheet_180328.pdf). We added this as a reference. The precision that the reviewer mention is also from the datasheet and can be regarded as an upper limit that is guaranteed by the manufacturer, so that a higher precision achieved in reality as shown in this work is possible.

Comment: Line 253. The DOAS method relies on cross-calibration with a Hantzsch. While two different Hantzsch instruments are used, is it possible that any systematic offsets related to the Hantzsch methodology could influence DOAS observations? I suggest showing the data used in the regression analysis discussed in lines 248-252.

Response: From our years-long experience in operating Hantzsch instruments we do not see any reason for a persistent systematic offset between two instruments. Both have been calibrated using the same procedure and the same commercially available calibration solutions, hence potential differences in their detection

sensitivities would be corrected for by the calibration. Showing the plot of the regression analysis between the DOAS and Hantzsch data, as suggested by the reviewer, would not be of help to detect a possible systematic difference between the two Hantzsch instruments.

Comment: Line 263-265. Again, it is unclear how the 6% accuracy is calculated. More details are needed. Does it take into account the uncertainty in the absorption cross section from the regression described in lines 248-252?

Response: The rotational-vibrational absorption lines of HCHO around 308 nm are very narrow (in the low picometer range) and consequently the experimentally observes cross section is strongly dependent on the spectral resolution of the instrument. Meller and Moortgat used a resolution of 0.025 nm while our DOAS instrument has a resolution of 0.0025 nm and therefore the cross sections are different. We changed the text L252: "The accuracy of the DOAS formaldehyde data is estimated to be 7%. It is mainly determined by the accuracy of the calibration procedure of the former Hantzsch instrument which was used for the comparison with the DOAS. It also takes into account the uncertainty in the absorption cross section from the regression between the DOAS and Hantzsch data which was 0.16%." We give additional explanation in L253: "The high-resolution cross section determined in this work compares very well with the value inferred from concurrent chamber measurements by a low-resolution DOAS and the high-resolution instrument by Brauers et al. 2007, which resulted in a differential cross section of $8.97 \times 10^{-21}$ cm^2, a value which is well within the stated accuracy of 7%. Therefore, DOAS measurements in the comparison here can be regarded as independent from Hantzsch measurements."

Comment: Figure 1. I suggest showing HCHO concentrations using each method in panel 2, rather than a second day showing the same thing as panel 1.

Response: In our opinion, plotting calculated mixing ratios of a zero signal does not illustrate exactly what we intend to show. The point we would like to make is that there is a changing zero signal that needs to be adequately monitored. Plotting mixing ratio would require subtracting the changing zero signal from the measurements and the information, how much the zero signal changes over time is lost. Therefore, we prefer keeping the plot as it is. The information how the signal converts to an equivalent HCHO concentration is given in the caption.

Comment: Line 291. If the Picarro specifies an anticipated zero drift of 1.5 ppb, I think the offset should not be classified as a "bias", but rather a zero-point that needs to be taken into account. The offset of the Hantzsch instrument is not called a "bias".

Response: We changed the wording as suggested by the reviewer.

Comment: Line 366: Is the intercept -0.13 ppb?

Response: Thanks for noticing the typo that we corrected in the revised version.

Comment: Lines 406-416. It is unclear how the main points of this paper can be applied to future deployments of the CRDS method. How often does a HCHO zero need to be taken with the CRDS instrument if ambient H2O varies? Could the HCHO zeroing system used for the Hantzsch be applied to the Picarro? Can the authors reassess their observations using the new software?

Response: Our observation demonstrate drifts are within the specification of the manufacturer, but that there is need for regular zero measurements, if the user wants to achieve a higher accuracy. From what we see, zeroing once a day would be sufficient and characterization of the humidity dependence on a regular basis. Our dataset does not allow to specify how often this needs to be done. Unfortunately, we could not install the new software version to test, if the water vapour dependence of the zero disappears. We added L425: "Observations in this work suggest that zero measurements should be done once a day and that the water dependence of the zero point of the CRDS instruments does likely not significantly change at least for a month-long deployment."

Comment: Throughout. The authors switch between 1 sigma and 3 sigma LODs. A consistent reporting methodology would improve the readability.

Response: All statements concerning the limit-of-detection are now given as 3- sigma LODs.

Response to the comments of referee #1:

We thank the reviewer for the helpful comments.

Comment: Line 65: LIF for HCHO does not require a custom fiber laser, e.g., St. Clair et al., 2019. https://doi.org/10.5194/amt-12-4581-2019

Response: We changed the text: "LIF instruments also custom-built and instruments often make use of a fibre laser for the excitation."

Comment: Line 83: "Comparisons" should not be capitalized.

Response: Done.

Comment: Line 211: The Russell et al., 2020 citation uses the Picarro instrument, but provides no understanding of the instrument itself and adds no value to this manuscript. Perhaps it can be considered an instrument intercomparison, but that seems a real stretch. Your work here is considerably better on that topic.

Response: We agree that the reference does not serve as a reference for understanding the instrument and cancelled it in L211.

Comment: Line 111: "Reports of instrument comparisons concluded that the measurement of formaldehyde remains challenging specifically for atmospheric concentrations in the low ppbv range." This may be true for commercially available instruments, but is not true for research-grade instruments. You should qualify this statement.

Response: We added at the end of the sentence "in particular for commercial instruments" to qualify the statement.

Comment: Line 253: "Therefore, DOAS measurements can be regarded as independent from Hantzsch measurements in the comparison here." I strongly disagree with this statement. The DOAS data is produced using an empirical calibration where a Hantzsch instrument was the standard. They each may provide some unique information for data evaluation, but they should not be considered independent measurements. On that point, I do not understand how the DOAS measurement can have a higher accuracy (6%) than the technique used to calibrate it (Hantzsch, 8.5%).

Response: We agree with the reviewer and deleted the statement in line 254. We corrected the stated accuracy of 6 % which misleadingly refers to the calibration uncertainty of the OH radical cross section and added the following text (L253): "The combined accuracy of the DOAS instrument is 7%. It is basically given by accuracy of the calibration procedure of the former Hantzsch instrument which was used for the comparison with the DOAS. The given accuary also takes into account the uncertainty in the absorption cross section from the regression between the DOAS and Hantzsch data which was 0.16%." We give additional explanation in the same paragraph: "The high-resolution cross section determined in this work compares very well with the value inferred from concurrent chamber measurements by a low-resolution DOAS and the high-resolution instrument by Brauers et al. 2007, which resulted in a differential cross section of 8.97x10^-21 cm^2, a value which is well within the stated accuracy of 7%. Therefore, DOAS measurements in the comparison here can be regarded as independent from Hantzsch measurements."

Comment: Section 3.2: How do you know that the size of the water-dependent offset determined at HCHO=0 is the same size as the water-dependent offset in the presence of HCHO? Since this is a spectroscopic interference, it seems possible that the fitting error caused by water will be dependent on the magnitude of the HCHO signal as well. A zeroing approach that removes HCHO but preserves humidity would be one solution. Is the 1.5 ppbv zero drift specified by Picarro due to this water interference or due to other factors?

Response: Indeed, there would be the chance that the water-dependent offset changes for higher HCHO. In this work, the water vapour dependence was determined, if no HCHO was present. This was then applied to all data. The correlation between measurements by the CRDS and the other instruments results in a good

agreement over a wide range of typical atmospheric concentrations. The level of agreement does not exhibit a water vapour dependence. This suggests that the water vapour dependence is applicable in the same way also for non-zero formaldehyde concentrations. The reviewer argues that the fitting error caused by water might also be dependent on the magnitude of the HCHO signal itself. Since details of the spectral evaluation, the modelling of the spectral overlap between the HCHO, H2O, and CH4 absorption lines, and the additionally applied empirical corrections are not publicly available this question is difficult to answer quantitatively. However, the poster presentation of Hoffnagle et al. at AGU 2017 can help. It shows the intensity of the overlapping absorption lines of CH4 @ 100 ppm, HO @ 2.2 %, and HCHO @7.4 ppm. From this one can estimate the HCHO line at a typical atmospheric mixing ratio of 5 ppb being about a factor of 6000 smaller than the water vapour line at 1 % water vapour. Taking this into account it seems unlikely that the presence of HCHO at atmospheric conditions would have an influence on the fitting error.

We added in the discussion of the correlation between measurements (L372): "This also demonstrates that the zero-point corrections determined can be applied over a wide range of atmospheric HCHO concentrations."

Comment: Figure 7 is rather small for the amount of data it contains. Please consider making it bigger.

Response: We scaled the figure to fit the width of one column in a final publication in AMT. We will pay again attention, if the figure is large enough, if the paper is type-set for final publication.

Response to the comments from Andrew Whitehill:

We thank Andrew Whitehill for the helpful comments.

Comment: Line 59: "Absorption spectroscopy has the advantage of being calibration-free" - this is highly misleading, as the Picarro instrument's formaldehyde retrieval is based partially on a factory-applied calibration and factory-programmed empirical correction factors. This is also contradicted by the later observation of a water vapor dependence on the accuracy of the instrument.

Response: Calibration-free does not mean that there is no careful characterization of the instrument's properties and/or zero measurements required, but that the concentration can be calculated without applying a sensitivity parameter that is derived from measurements using a calibration standard. We added in L59: "Absorption spectroscopy has the advantage that it does not need regular calibrations with a gas standard. But it requires knowledge of absorption cross sections and careful characterization of instrumental properties to avoid or correct possible spectral interferences and signal offsets."

Comment: Line 62: There are at least 4 commercially available spectroscopic formaldehyde instruments with sub-ppb sensitivity of which we are aware: Aerodyne Research (absorption spectroscopy at ca. 1765 cm-1) Aeris Technologies (absorption spectroscopy at ca. 2832 cm-1) Gasera (Gasera One Formaldehyde, photoaccoustic spectroscopy) Picarro (G2307, cavity ringdown spectroscopy at 5625.85 cm-1)

Response: Thanks for the hints. We added all instruments that are not mentioned yet in Line 62: "While in the past most of the instruments were custom-built and required substantial pre-knowledge of operators, commercial instruments based on absorption spectroscopic methods have recently become available (CRDS by Picarro Inc., TDLS by Aeros Technology and by Aerodyne Research, photoacustic spectroscopy by Gasera).

Comment: Section 2.3: Authors should expand their discussion of how they calibrated the Picarro instrument.

Response: In fact, there was no calibration done in the sense of providing standard additions of HCHO. To better explain, how the zero-point measurements were done, we added L 230: "For determining the instrumental zero in this work, measurements in the chamber are used, when no formaldehyde was present. The humidification process of the air in the chamber allowed to characterize the dependence of the instrumental zero on water vapour (Section 3.2)."

Comment: Line 223: Authors state that the accuracy of the Picarro G2307 is 10%. However, their own data (including the water vapor interference and the zero drift) contradict this 10% value, especially for low formaldehyde concentrations.

Response: The 10% accuracy is taken from the datasheet provided by Picarro (https://www.picarro.com/sites/default/files/Picarro_G2307%20Datasheet_180328.pdf). Indeed, the accuracy could be worse, if the water vapor correction is not applied as shown in this work.

Comment: Section 3.2 - Authors should make it more clear when they are discussing synthetic (methane-free) air and when they are discussing air with ambient methane concentrations. In particular, authors ignore the potential methane (and methane+water) cross-sensitivities of the instrument and should address his in their discussion and analysis of data.

Response: Thanks for the hint. We are aware of that there is an interference from methane in the CRDS measurements and observed that also in chamber experiments with exceptionally high methane concentrations. However, for ambient concentrations, this interference is rather small and does not significantly contribute to the signal. To clarify this point, we added L 324: "In experiments with exceptionally high non-atmospheric mixing ratios of methane, also an offset appears that depends on methane. However, for atmospheric mixing ratios as experienced in this work, this additional offset was not significant and therefore it is neglected."